# Isotopic evidence for geographic heterogeneity in Ancient Greek military forces

Katherine L. Reinberger[1]*, Laurie J. Reitsema[1], Britney Kyle[2], Stefano Vassallo[3], George Kamenov[4], John Krigbaum[5]

1 Department of Anthropology, University of Georgia, Athens, Georgia, United States of America, 2 Department of Anthropology, University of Northern Colorado, Greeley, Colorado, United States of America, 3 Soprintendenza BB.CC.AA. di Palermo, Palermo, Italy, 4 Department of Geological Sciences, University of Florida, Gainesville, Florida, United States of America, 5 Department of Anthropology, University of Florida, Gainesville, Florida, United States of America

* klreinberger@uga.edu

**Data Availability Statement:** All relevant data are within the paper.

**Funding:** Data collection in Sicily, Italy and strontium isotope analyses was funded by a

## Abstract

Increased mobility and human interactions in the Mediterranean region during the eighth through fifth centuries BCE resulted in heterogeneous communities held together by political and cultural affiliations, periodically engaged in military conflict. Ancient historians write of alliances that aided the Greek Sicilian colony Himera in victory against a Carthaginian army of hired foreign mercenaries in 480 BCE, and the demise of Himera when it fought Carthage again in 409 BCE, this time unaided. Archaeological human remains from the Battles of Himera provide unique opportunities to test early written history by geochemically assessing the geographic origins of ancient Greek fighting forces. We report strontium and oxygen isotope ratios of tooth enamel from 62 Greek soldiers to evaluate the historically-based hypothesis that a coalition of Greek allies saved Himera in 480 BCE, but not in 409 BCE. Among the burials of 480 BCE, approximately two-thirds of the individuals are non-local, whereas among the burials of 409 BCE, only one-quarter are non-local, in support of historical accounts. Although historical accounts specifically mention Sicilian Greek allies aiding Himera, isotopic values of many of the 480 BCE non-locals are consistent with geographic regions beyond Sicily, suggesting Greek tyrants hired foreign mercenaries from more distant places. We describe how the presence of mercenary soldiers confronts prevailing interpretations of traditional Greek values and society. Greek fighting forces reflect the interconnectedness and heterogeneity of communities of the time, rather than culturally similar groups of neighbors fighting for a common cause, unified by "Greekness," as promoted in ancient texts.

## Introduction

Human mobility played a central role in colonization, ethnogenesis, and warfare in the ancient Mediterranean region [1–4]. Ongoing clashes in the 8th-5th centuries BCE between indigenous groups, Greeks, Persians, and Phoenicians created unprecedented instances of interpersonal

Research Experience for Undergraduates from the National Science Foundation awarded to LJR and BK (https://www.nsf.gov/funding/pgm_summ.jsp?pims_id=5517), award numbers 1560227 and 1560158. Oxygen isotope analyses were funded by the University of Georgia Graduate School Innovative and Interdisciplinary Research Grant and Dean's Award, the Willson Center for Humanities and Arts Graduate Research Award, and the UGA Center for Archaeological Science Norman Herz Grant for Student Research, awarded to KLR. The funders had no role in study design, data collection and analysis, decision to publish, or preparation of the manuscript.

**Competing interests:** The authors have declared that no competing interests exist.

interaction, violent or otherwise. Antagonistic interactions are not just a reflection of people's perceptions of each other and themselves, but also a stimulus for the formation of new ethnic identities and political forms, bringing some groups together while cleaving others [2, 5, 6]. To examine the role military activities played in bringing together diverse communities of people in the ancient world, we report the strontium and oxygen isotope ratios of skeletons from mass graves associated with two historically significant and violent conflicts–the Battles of Himera in 480 BCE and 409 BCE between Punic Carthage and Sicilian Greeks at the Greek colony, Himera.

Traditional interpretations of ancient military construction have relied on historical and literary narratives by ancient authors such as Aristotle, Thucydides, and Homer. *The Histories* by Herodotus, written around 440 BCE and regarded as the earliest work of history in western literature, was largely an account of clashes among ancient peoples, focusing particularly on the Persian War, which occurred during Herodotus' lifetime. Such ancient histories are invaluable for our recognition and descriptions of how people and societies are formed and function today and in the past, but also are incomplete and carry the biases of individual authors [7]. For example, writing at a time of ongoing warfare, Herodotus' descriptions augment a distinction between Greeks and "barbarians," deliberately glorifying Greeks citizenry [8, 9].

We use strontium ($^{87}Sr/^{86}Sr$) and oxygen ($\delta^{18}O$) isotope ratios of human tooth enamel to explore the geographic place of origins of participants in the two Battles of Himera. Using this direct evidence, we test two hypotheses derived from historical records of each battle: (1) the battle in 480 BCE was fought by allied combatants assembled from other Greek cities on Sicily (Diod. 11.20–21; Hdt. 7.165–7), and (2) the second battle in 409 BCE was fought chiefly by Himerans themselves (Diod. 13.62). Describing the geographic points of origins of Greek fighting forces helps situate the role of political factors in Sicily, especially how Greek tyrants may have influenced military activities and utilized the interconnectedness of the Mediterranean world to recruit soldiers from beyond local populations.

## Military forces and Sicilian Greek city-states

Geopolitically, during the Archaic and Classical periods "Greece" was a series of city-states with varied dialects, traditions, and ethnicities. Greeks tended to divide the world into Greeks and barbarians, where in the Hellenic mindset, non-Greeks were considered barbarians [10]. "Greekness" was defined by one's ability to trace their origins to Greece and would have been an important factor in determining citizenship within the *poleis* (city-states). During the Archaic and Classical periods, armed Greek forces chiefly comprised hoplite soldiers: heavily armored, self-funded citizens who served their own *poleis* and allies. Most hoplites in Classical mainland Greece would have been part of small militias of neighbors and relatives defending their land, rather than professional, paid soldiers [11, 12]. Hoplites have been seen as the archetype of Greek ideals of honor, citizenship and democracy (Arist. Nic. Ethics 3 1116b15) [12, 13].

Mercenaries, or paid soldiers, were used in foreign armies across the Mediterranean especially after the spread of coinage in the Archaic period (Hdt 7.165) [14, 15]. Mercenaries are known to have aided both Greek *poleis* [13] and foreign polities (helping Persia; Hdt. VIll.26.52; Thuc.I.60.). Early mercenaries (8th-7th centuries BCE) were often small groups of specialists brought in to support citizen-based armies, and could be of Greek or other cultural backgrounds [13]. Some of these specialized troops may have been intentionally recruited from areas known for their military prowess, such as archers from foreign Scythia [12].

Sicily's military practices and political organization diverge from mainland Greece in the predominance of tyrannies, with only small interludes of democracy [16]. Tyrants were city

leaders not necessarily from the cities themselves who rose to power by populism or force. While some tyrants were supported by the populace, others were not, and had to work strategically to maintain tenuous control [17, 18]. Tyrants were known to forcefully relocate segments of the populace, and hire bodyguards for protection, including foreign mercenaries (Diod. 11.49; Hdt. 7.156; Thuc. 6.4.6) [16, 19]. Importantly, Greek tyrants pursued territorial expansions across Sicily, leading to the division of the island into pro-Carthaginian (Selinunte, Himera), and anti-Carthaginian (Syracuse, Agrigento) tyrannies [20].

## Site description and the Battles of Himera (480 BCE, 409 BCE)

The Greek city of Himera was founded on Sicily's northern coast around 648 BCE by culturally Greek settlers from the colony of Zankle, modern day Messina, and political exiles from Syracuse (Thuc 6.5.1) [21]. Due to its strategic location (Fig 1), Himera commanded the sea routes along Sicily's Tyrrhenian coast, as well as major land routes across the island. Himera was divided into an upper city, with evidence of agricultural processing and stables [22] and a lower city, housing merchants and craftsmen with ready access to the port and *agora* [19, 22].

While the first Battle of Himera was a Punic attack of Himera, it is more appropriately thought of as the culmination of multiple political skirmishes and alliances between the tyrants of Agrigento, Syracuse, and Himera, and of Sicilian Greek *poleis*' differing relationships to the Phoenician world, ranging from friendly to hostile [23]. Political unrest in Sicily inspired Carthage to attack Himera in 480 BCE (Hdt. 7.165) and Gelon, the tyrant ruling Syracuse, came to

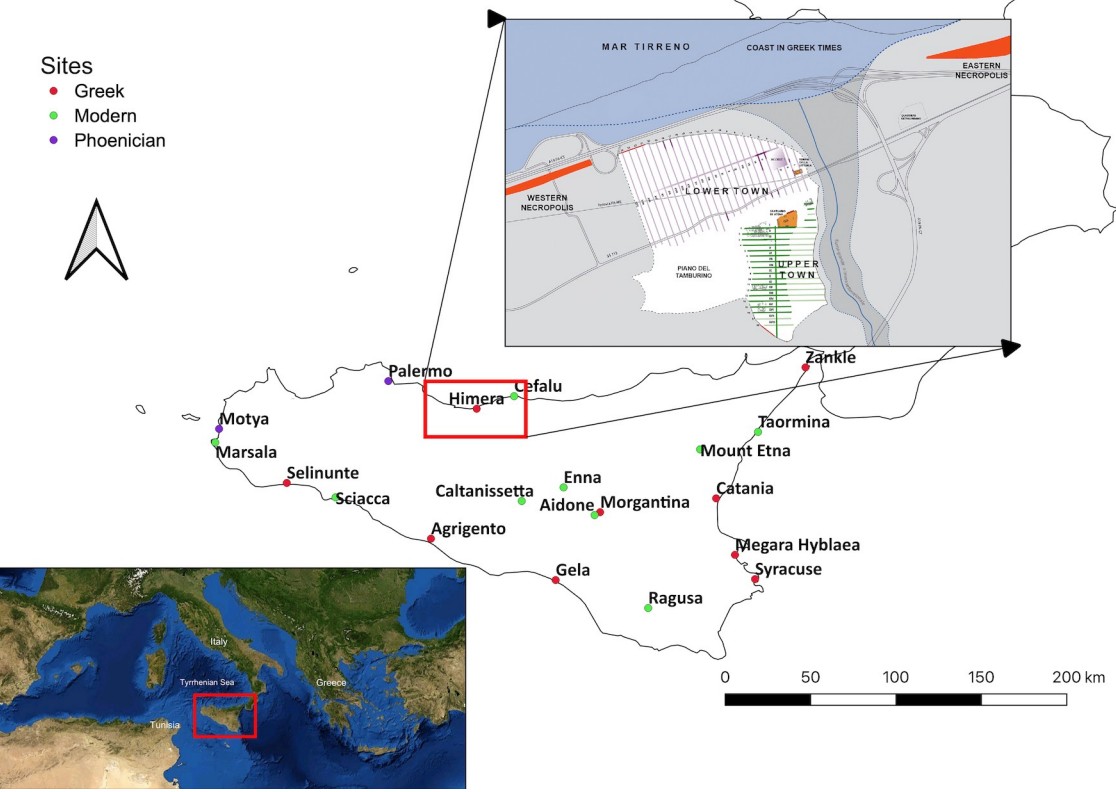

**Fig 1. Map of Sicily showing Greek and Phoenician colonies in the 5[th] century in the context of modern towns.** The plan of Himera shows the relation of the Upper and Lower towns to the river and the Western necropolis where the Battles of Himera were fought along the western fortifications. Public domain image of the Mediterranean Sea via Wikimedia Commons from NASA World Wind, modified to show location of Sicily. Map of sites in Sicily created in QGIS by KLR and plan of Himera created by SV.

Himera's aid to protect his economic and political interests (Diod. 11.20). The Battle of 409 BCE at Himera continued the conflict between Greek hegemony and the personal and economic interests of Punic rulers who had family connections to soldiers killed in 480 BCE (Diod. 13.61). At the beginning of the battle, allied forces from Agrigento and Syracuse rallied to assist Himera, but upon perceiving the threat Carthage posed to their own cities, they departed, evacuating many Himeran citizens in the process as they left (Diod. 13.61). Unaided, relying on all available men, young and old, and fighting "with no thought for their lives" (Diod. 13.61–62) Himera fell and the Carthaginians razed the city and killed remaining citizens who had not already fled.

## Ancient authors and the discovery of the mass graves

Much of our knowledge of the Battles of Himera is from the historical texts of Herodotus and Diodorus Siculus. Herodotus (c. 484–425/413 BCE) lived in Asia Minor and documented his travels during the Persian Wars and the first Battle of Himera. Herodotus emphasizes the Greek-barbarian dichotomy, often focusing only on conflicts between these groups, while also being wary of empires and tyrants, and completely against wage-earning mercenaries [8, 9]. Diodorus Siculus lived in the 1st century BCE Roman period in Sicily. Although he is credited with putting together an immense composite history of the Mediterranean world, his work is criticized for copying what other historians already had written, and for amplifying Sicily's importance [23]. Both Herodotus and Diodorus Siculus assert and emphasize that the first battle in 480 BCE occurred on the same day as other Greek victories in the Battle of Thermopylae and the Battle of Salamis "as though heaven had deliberately arranged for the finest victory and the most famous of defeats to take place simultaneously" (Diod. 11.24, translated by Green 2010; Hdt. 7.166) [20], a possible exaggeration of the exploits of the colonial Greek West [20, 23]. The discovery of archaeological evidence of the battles at Himera, including military arms and the mass graves of soldiers, has allowed researchers to evaluate these historical texts and their authors.

Archaeological excavations at Himera revealed eight mass graves containing 132 individuals, all adult males [24]. Seven graves date to 480 BCE (FC, *fossa comune*, 1–7) and one to 409 BCE (FC 8+9, originally numbered as separate mass graves but now thought to represent a single mass grave [25]) based on associated pottery and stratigraphy [22]. Their location on the documented battlefield site, stratigraphic position, dates, adult age, osteological male sex identification [26], and presence of weapons and violent trauma on several of the skeletons suggest they were soldiers [22, 24, 25, 27].

That these soldiers fought on behalf of Himera is suggested by their intentional burial [22], implying burial by Greek victors who had time and opportunity to respectfully bury their own dead. The 480 BCE mass graves show orderly layouts of bodies, in rows, side by side, with heads facing east. In slight contrast, the mass grave from 409 BCE contains many more individuals but in two levels, and while the lower layer of bodies is orderly (bodies side by side, head-to-head), a stacked second layer of burials is crowded, with many bodies positioned head-to-foot [24].

Beyond the descriptions of ancient authors and archaeological contextual evidence suggesting the individuals from the mass graves fought for Himera, little is known about the soldiers' origins. Isotopic analysis of the skeletons themselves provides a needed direct line of evidence complementary to the written sources.

## $^{87}Sr/^{86}Sr$ and $\delta^{18}O$ isotopic analysis

Isotopic analysis of human bones and teeth offers a direct, individualized window into past migration and diet [28]. Strontium isotopes ($^{87}Sr/^{86}Sr$) are incorporated into bones and teeth

through the consumption of food and water that reflect the isotopic ratios of the soil and bedrock of the region where an individual lived at the time of tissue formation [29]. Strontium substitutes for calcium in the mineral component of teeth and bone, phosphate hydroxyapatite $(Ca_{10}(PO_4)_6(OH)_2)$, because they are both alkaline earth elements with a valence of +2 and similar ionic radii [30–32].

There are four stable isotopes of strontium in nature, $^{88}Sr$ (82.53%), $^{87}Sr$ (7.04%), $^{86}Sr$ (9.87%), and $^{84}Sr$ (0.56%) [33]. The small mass differences between the isotopes of strontium results in negligible fractionation as the elements are incorporated into the hydroxyapatite of bones and teeth.

$^{87}Sr$ is radiogenic and comes from the radioactive decay of $^{87}Rb$. Therefore, the present-day $^{87}Sr/^{86}Sr$ ratio in a rock is a function of the initial $^{87}Sr/^{86}Sr$ ratio, rubidium and strontium content (Rb/Sr), and the age of the rock [31]. For example, continental rocks have a greater range of Sr isotopic variation than oceanic basalts [34]. Rocks such as shales and granites tend to have high Rb/Sr and high $^{87}Sr/^{86}Sr$ ratios (typically above 0.710) [35]. In contrast, volcanic rocks such as basalts and andesites have low Rb/Sr and typically low $^{87}Sr/^{86}Sr$ ratios (less than 0.706 [35]). Additionally, coastal environments sometimes can be discerned via their higher values caused by sea spray and exposure to the constant $^{87}Sr/^{86}Sr$ value of the ocean (0.7092) [36]. Strontium isotopes are incorporated into the food chain from the rocks, to soil and groundwater, and then are made bioavailable to heterotrophs via plants [31].

Whether the $^{87}Sr/^{86}Sr$ values of humans are determined to be local or non-local depends on the $^{87}Sr/^{86}Sr$ values of the local environment, which can be assessed by measuring modern fauna, archaeological fauna, and soils [31]. The geology of Sicily is dominated by Meso-Cenozoic formations (Triassic-Tertiary deposits, Fig 2), mostly composed of carbonate and siliciclastic sediments [37]. Limited outcrops of older Paleozoic and Precambrian rocks are exposed in the northeastern tip of the island (Fig 2). Younger Quaternary volcanic rocks are associated with the Mount Etna volcano in the eastern part of the island [29, 37]. The area around Himera is characterized by a range of deposits from Upper Mesozoic to Late Miocene age. Agrigento has slightly younger deposits, comprising the Late Miocene to the Pleistocene [37]. Syracuse is

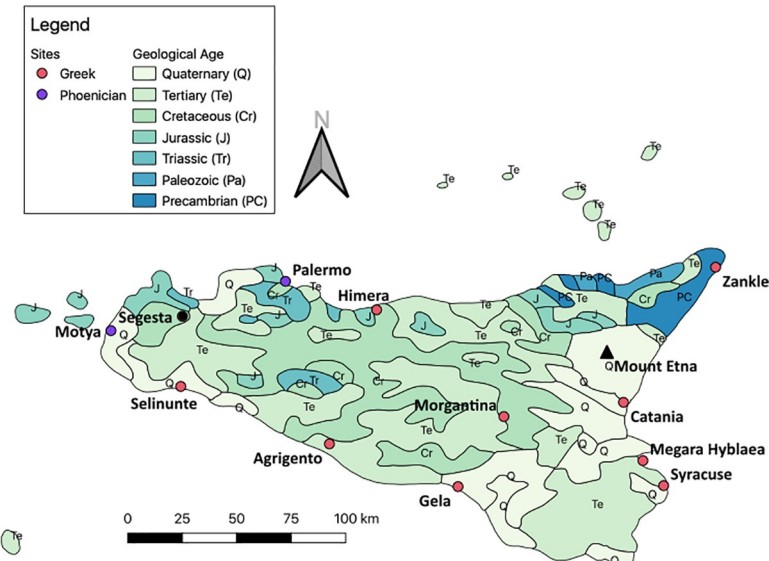

**Fig 2. Map depicting geologic ages of the underlying bedrock in Sicily.** Youngest deposits are represented by the lightest color and get darker with older geologic ages. Created in QGIS using DataSource: USGS [39] by KLR.

located on the Hyblean Plateau. The plateau bedrock is dominated by Cenozoic carbonate rocks, ranging in age from Oligocene-Miocene to Pliocene-Pleistocene [38].

Unlike bones, teeth do not remodel over time. Teeth preserve the $^{87}Sr/^{86}Sr$ ratios of a person's environment during childhood and are widely used to track ancient human migration and movements [40–45].

Oxygen isotope ratios ($\delta^{18}O$) provide complementary insights into mobility and migration [42, 46–49]. Oxygen is incorporated into bone and enamel hydroxyapatite primarily through drinking water. The $\delta^{18}O$ values of drinking water are influenced by environmental factors such as temperature, humidity, altitude, and latitude [50, 51]. Fig 3 is a spatial interpolation of estimated $\delta^{18}O$ values for Sicily, based on calculated values from the Online Isotopes in Precipitation Calculator (OIPC: http://wateriso.utah.edu/waterisotopes) [52]. Previous research in the ancient Mediterranean region has shown patterned variations in $\delta^{18}O$ values of human tooth enamel associated with latitude and proximity to coasts [46, 47, 53, 54].

Today, the annual precipitation rate in Sicily is 500–600 mm, with Agrigento averaging around 500mm, Himera at 550 mm, and Syracuse at 590 mm [57]. The mean annual temperature at Himera (15.9° C) is slightly lower than Agrigento (17.3° C) and Syracuse (17.6° C). This suggests that $\delta^{18}O$ will not differ dramatically between the sites, though Syracuse may have slightly higher $\delta^{18}O$ values than Himera because of higher temperatures, and Agrigento may have lower $\delta^{18}O$ values, being situated at a higher altitude (~84 m versus Syracuse at ~15 m, Himera's Upper City at 72 m and Lower City at 9 m). The civilians living in the upper city

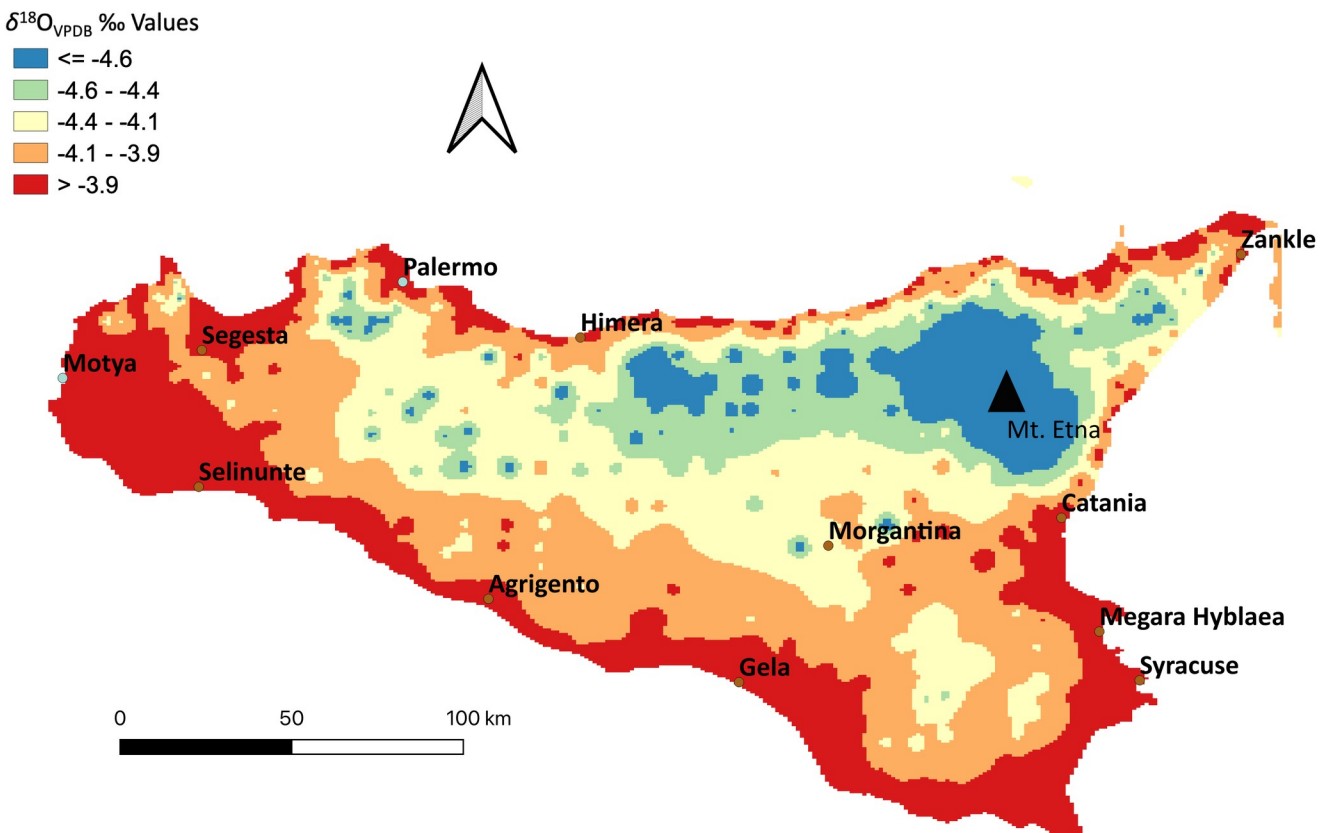

**Fig 3. Map of predicted $\delta^{18}O_{VPDB}$ variability on Sicily.** An Inverse Distance Weighting (IDW) Interpolation was created in QGIS from collected modern water, modern teeth, and calculated values from OIPC [52]. Values from OIPC were calculated by inputting longitude, latitude, and altitude of locations across Sicily. Values from modern water and OIPC were converted from $\delta^{18}O_{VSMOW}$ to $\delta^{18}O_{VPDB}$ ‰ [55, 56].

of Himera likely consumed water that was stored from retained rainwater in cisterns, while those living in the lower city had more access to the river, in addition to artesian wells to collect from the local aquifer [19, 58]. These different sources may result in slightly lower $\delta^{18}$O values in the upper city because cisterns were less susceptible to evaporation.

## Materials and methods

### Materials

We analyzed $^{87}$Sr/$^{86}$Sr and $\delta^{18}$O values of tooth enamel from 62 individuals interred in eight mass graves associated with the Battles of Himera, including 51 from 480 BCE and 11 from 409 BCE. Due to poor preservation, this sample represents 100% of the mass grave individuals having at least one tooth present for analysis. We also included 25 adult individuals (13 males, 9 females, 3 indeterminate) from the surrounding cemetery to represent the contemporaneous, general population of Himera. All individuals are from Himera's western necropolis [19]. Enamel from second molars which form during 2–8 years of age are preferentially used, followed by premolars which form during 1–8 years of age [59]. Therefore, $^{87}$Sr/$^{86}$Sr and $\delta^{18}$O values reflect the local geology and water sources, respectively, where individuals lived when they were between 1 and 8 years of age [60].

In order to identify possible locals and non-locals, it is necessary to establish a range of isotope values local to Himera. For the local $^{87}$Sr/$^{86}$Sr baseline, we analyze modern fauna spotted on the ground at and near the archaeological site of Himera, archaeological fauna discovered occasionally in the burial boxes during osteological analysis, modern permanent human teeth donated by local dentists' offices in nearby Campofelice di Roccella, and soils recovered from inside archaeological human bone cortices (Table 1). At Agrigento we analyze modern land snails, faunal bones, and soil (Table 2). The baseline for Syracuse was estimated based on the local geology of the Hyblean Plateau.

To evaluate whether soldiers were derived from Himera's general populace, we use a Mann-Whitney U test to compare soldiers to the mean plus and minus one standard deviation of 25 individuals from Himera's western necropolis not associated with mass graves. This estimation of the general populace is not the same as a "local baseline" because members of the general populace, which as a colony, may also have been raised elsewhere, but is necessary to evaluate whether soldiers do or do not represent the general populace in terms of diversity in geographic origins. To estimate the local, environmental baseline for Himera, we consider the average annual $\delta^{18}$O value for precipitation of -5.4‰ from OIPC, and we report $\delta^{18}$O measurements of (1) modern waters from taps and fountains from areas near Himera collected in 2017–2018 (Table 1), and (2) modern tooth enamel from a dentist's office in Campofelice di Roccella in 2018 (Table 2). Oxygen isotope environmental baselines are estimated for Agrigento and Syracuse using OIPC estimated precipitation values (-5.8‰ for Agrigento and -5.3‰ for Syracuse), and measurements of local waters from taps and fountains collected in 2017–2018 (Table 2).

Specimens are stored at the Parco Archeologico di Himera, Sicily, Italy. Permits for the collection of samples for transportation to the Bioarchaeology and Biochemistry Laboratory at the University of Georgia, Athens, GA, USA and isotopic analysis were issued by the Dipartimento dei Beni Culturali e dell'Identità Siciliana and the Soprintendenza ai Beni Culturali e Ambientali di Palermo. All necessary permits were obtained for the described study, which complied with all relevant regulations.

### Methods

Tooth enamel samples were prepared at the University of Georgia Bioarchaeology and Biochemistry Laboratory. Approximately 60mg of tooth enamel was cleaned to remove external

**Table 1. Himera isotopic baseline.**

| ID | Sample Type–Source | $\delta^{18}O_{water}$ (VSMOW, ‰) | $\delta^{18}O_{carbonate}$ (VPDB, ‰) | $^{87}Sr/^{86}Sr$ |
|---|---|---|---|---|
| CF-TEETH-1 | Modern enamel | | -4.3 | 0.70897 |
| CF-TEETH-2 | Modern enamel | | -4.0 | 0.70970 |
| CF-TEETH-3 | Modern enamel | | -4.4 | 0.70858 |
| CF-TEETH-4 | Modern enamel | | -4.1 | 0.70908 |
| H1-18 | Modern water–Sink tap | -6.4 | -4.2* | |
| H2-18 | Modern water–Sink tap | -6.5 | -4.3* | |
| H3-18 | Modern water–River | -7.1 | -4.6* | |
| H-Upper City | OIPC | -5.5 | -3.7* | |
| H-Lower City | OIPC | -5.3 | -3.6* | |
| H-W1993 | Arch. Fauna–Dog | | | 0.70878 |
| H-W1114-P | Arch. Fauna–Pig | | | 0.70896 |
| HMFCAP | Arch. Fauna–Caprine | | | 0.70863 |
| H-W3704-D | Arch. Fauna–Dog | | | 0.70875 |
| H-W704-H | Arch. Fauna–Horse | | | 0.70913+ |
| H-W-3030-H | Arch. Fauna–Horse | | | 0.70882+ |
| H-F-L1 | Modern Fauna | | | 0.70846 |
| H-F-R1 | Modern Fauna | | | 0.70871 |
| H-F-R2 | Modern Fauna | | | 0.70841 |
| H-F-R3 | Modern Fauna | | | 0.70839 |
| H-F-M1 | Modern Fauna | | | 0.70866 |
| H-Alta-N-SH | Snail shell | | | 0.70858 |
| H-Alta-T-SH | Snail shell | | | 0.70864 |
| HSH-1 | Snail shell | | | 0.70849 |
| HSH-2 | Snail shell | | | 0.70853 |
| HSH-3 | Snail shell | | | 0.70825 |
| H-W234-S | Soil | | | 0.70859 |
| H-W699-S | Soil | | | 0.70857 |

*$\delta^{18}O$ values were calculated from Coplen 1988 and Chenery 2012 equations [55, 56].

+Values from horses were not used to calculate the baseline

surfaces and adhering dentin using a Dremel® hand-held rotary tool and a diamond-embedded drill bit. Cleaned tooth enamel was then broken into ~1mm sized fragments in a steel mortar and pestle Each sample was divided by mass into two roughly equal subsamples for oxygen analysis and strontium analysis. Subsamples for strontium isotope analysis were sent to the University of Florida Bone Chemistry Lab in the Department of Anthropology where human and faunal samples were inventoried, assessed, and prepared for ion chromatography. Modern snail shell samples, collected in the field, were mechanically cleaned and sonicated in deionized-distilled water to remove exogenous contaminants.

All samples were then processed in a class 1000 clean lab, equipped with class 10 laminar flow hoods in the University of Florida's Department of Geological Sciences. Human and faunal enamel samples were weighed into precleaned Teflon vials and dissolved in 3 ml 50% nitric acid ($HNO_3$) and evaporated to dryness on a 120˚ C hotplate under laminar flow. Modern snail shell were also weighed into precleaned Teflon vials, but first oxidized with 2 ml of 30% hydrogen peroxide ($H_2O_2$) for 24 h and then rinsed to neutral. Snail shells were then acidified with 0.5 ml of 0.5% acetic acid ($CH_3COOH$), and then 1 ml of 4x $H_2O$ and 0.5 ml of 50% $HNO_3$ was added to dissolve each sample, and then, once dissolved, samples were evaporated

**Table 2. Agrigento (AG) and Syracuse (SY) isotopic baseline.**

| ID | Sample Type—Source | $\delta^{18}O_{water}$ (VSMOW, ‰) | $\delta^{18}O_{carbonate}$ (VPDB, ‰) | $^{87}Sr/^{86}Sr$ |
|---|---|---|---|---|
| AG1-18 | Modern water–Sink tap | -6.5 | -4.3* | |
| AG2-18 | Modern water–Sink tap | -6.5 | -4.3* | |
| AG3-18 | Modern water–Sink tap | -6.6 | -4.3* | |
| AG4-18 | Modern water–Sink tap | -6.6 | -4.4* | |
| AG5-18 | Modern water–Spigot | -6.4 | -4.3* | |
| AG1-17 | Modern water–Sink tap | -7.4 | -4.9* | |
| AG1-17 | Modern water–Spigot | -6.7 | -4.4* | |
| AG-Valley | OIPC | -5.4 | -3.6* | |
| AG-City | OIPC | -5.8 | -3.9* | |
| AG-CA | Modern fauna | | | 0.70899 |
| AG-F-P | Modern fauna | | | 0.70892 |
| AG-MU-SH1 | Modern shell | | | 0.70892 |
| AG-MU-SH2 | Modern shell | | | 0.70897 |
| AG-MU-SH3 | Modern shell | | | 0.70899 |
| AG-MU-SH4 | Modern shell | | | 0.70900 |
| AG-TE-SH | Modern shell | | | 0.70891 |
| SY1-18 | Modern water–Sink tap | -5.1 | -3.5* | |
| SY2-18 | Modern water–Spigot | -5.1 | -3.4* | |
| SY3-18 | Modern water–Drinking fountain | -5.2 | -3.5* | |
| SY4-18 | Modern water–Sink tap | -5.0 | -3.4* | |
| SY-Ortigia | OIPC | -5.3 | -3.6* | |

*$\delta^{18}O$ values were calculated from Coplen 1988 and Chenery 2012 equations [55, 56].

to dryness at 120˚ C under laminar flow. Soil samples were leached using 2N HCl, with each acid leachate pipetted off to capture the bioavailable fraction of Sr. All soil leachates were dissolved in pre-cleaned Teflon vials using 8N $HNO_3$, after which Sr was separated from single aliquots using ion chromatography.

For all samples, strontium was separated by ion chromatography from single aliquots, using a selective resin that absorbs strontium. The dried residues were dissolved in 3.5N $HNO_3$ and loaded onto cation exchange columns packed with strontium-selective crown ether resin (Sr-spec, Eichrom Technologies, Inc.) to separate Sr from other ions [61]. Each 100 $\mu$l column stem was packed with Sr-spec resin, washed with 2 ml 4x$H_2O$ and equilibrated with 2 ml 3.5N $HNO_3$ (Optima). Dissolved samples were loaded onto the resin columns and washed four times with 100 $\mu$l 3.5N $HNO_3$ (Optima), then washed with 1 ml 3.5N $HNO_3$. Strontium was collected in 1.5 ml 4x$H_2O$ and evaporated to dryness on a 120˚ C hot plate under laminar flow. $^{87}Sr/^{86}Sr$ values were measured using a Nu-Plasma multiple-collector inductively-coupled-plasma mass spectrometer (MC-ICP-MS) using time-resolved analysis (TRA) [62]. The reported $^{87}Sr/^{86}Sr$ ratios are relative to NBS 987 $^{87}Sr/^{86}Sr$ = 0.710246 (± 0.00003, 2σ).

Subsamples for $\delta^{18}O$ analysis were further crushed to a coarse powder and treated with sodium hypochlorite (NaOCl) for 24 h to remove organic contaminants, and then with acetic acid (4h) to remove exogenous carbonates, following [63]. At the University of Georgia Center for Applied Isotope Studies, pretreated enamel powder was digested in 100% phosphoric acid and measured on a Thermo Scientific Gas Bench II coupled with a Thermo Scientific Delta V plus Isotope Ratio Mass Spectrometer (IRMS). $\delta^{18}O$ are reported relative to the VPDB carbonate standard and are expressed in per mil (‰) values. The mean $\delta^{18}O$ of Fisher analytical standard analyzed during sample runs is -14.9‰, with a standard deviation of 0.1‰.

Modern water samples were analyzed using a Thermo Thermal Conversion Elemental Analyzer connected to a Thermo Delta V plus IRMS at the Center for Applied Isotope Studies at the University of Georgia, and both VSMOW and VPDB values are provided in Tables 1 and 2. Oxygen isotope ratios were first converted to enamel carbonate (VSMOW) based on known water values: $\delta^{18}O_{VSMOW}$ Drinking water = 1.590($\delta^{18}O_{VSMOW}$ carbonate)– 48.634 [56]. These predicted enamel values were then converted to VPDB: $\delta^{18}O_{VPDB}$ = 0.97001($\delta^{18}O_{VSMOW}$) - 29.99 [55].

Data were analyzed using the statistical software platform R. Mann-Whitney U non-parametric tests were used for pair-wise comparisons. Non-parametric tests are appropriate for samples whose distributions deviate from normality and when sample sizes among groups differ. Levene's Test was used to assess equality of variances. Results were considered significant if p-values were less than 0.01.

# Results

## Baseline

A local environmental strontium baseline is calculated using the mean ±1 standard deviation of the mean of all archaeological fauna, modern land snails associated with the necropolis, and soil samples from the site. Values for the baseline are in Table 1. The calculated local range for $^{87}Sr/^{86}Sr$ is 0.70837–0.70900.

The $\delta^{18}O$ and $^{87}Sr/^{86}Sr$ values for the general populace and for individuals from both battles are presented in Table 3. Himera's $\delta^{18}O$ baseline is calculated using the mean ±1 standard deviation of 25 individuals from the general populace ($\delta^{18}O$ = -3.5±0.9‰). Local water and predicted precipitation values from Himera, Agrigento, and Syracuse were converted to predicted enamel values (VPDB) using previously established equations [55, 56]. All $\delta^{18}O$ values are reported in VPDB in the following interpretations.

The $\delta^{18}O$ range of individuals not interred in mass graves (general populace) is -4.4‰ to -2.5‰ (Table 3). This range corresponds well with the environmental baseline calculated from local water, predicted precipitation, and modern enamel from Campofelice di Roccella (-4.5‰ to -3.8‰) (Table 1). Sixteen percent of the individuals representing the general populace are estimated to be non-local to Himera. We use the general populace $\delta^{18}O$ range to estimate if individuals in the mass graves were likely local to Himera.

The strontium isotope baseline range for Agrigento is 0.70892–0.70900. The estimated range from local geology for Syracuse is 0.70800–0.70900, based on expected Sr isotopic compositions of Cenozoic (Oligocene to Pleistocene) carbonate rocks [64]. Based on precipitation estimates from OIPC and collected water samples, the $\delta^{18}O$ baseline for Agrigento is -5.1‰ to -3.1‰ and Syracuse is -4.5‰ to -2.5‰ (Table 2).

## 480 BCE

Individuals from the 480 BCE mass graves exhibit average $^{87}Sr/^{86}Sr$ = 0.709257 ± 0.00058 and are significantly different from the general populace (Mann Whitney U: W = 1033, p-value = 1.3x10$^{-5}$). Sixty-seven percent of the 480 BCE individuals are outside the local Sr range suggesting non-local origin. Most of the non-local individuals exhibit $^{87}Sr/^{86}Sr$ values higher than the local baselines for Himera, Agrigento, and Syracuse.

Individuals from the 480 BCE mass graves exhibit a mean $\delta^{18}O_{VPDB}$ value of -5.6±1.0‰. Based on the $\delta^{18}O_{VPDB}$ isotope values eighty-four percent of soldiers from 480 BCE are considered non-local. All non-local individuals exhibit $\delta^{18}O$ values below the expected baseline ranges for Himera, Agrigento, and Syracuse. The baseline ranges are shown with the results in Fig 4.

**Table 3. Study data table.**

| ID | Burial Type | $\delta^{18}$O (VPDB, ‰) | $^{87}$Sr/$^{86}$Sr |
|---|---|---|---|
| W1656 | Single inhumation | -2.4 | 0.70897 |
| W2574 | Single inhumation | -2.4 | 0.70855 |
| W1480 | Single inhumation | -2.4 | 0.70890 |
| W1788 | Single inhumation | -2.1 | 0.70871 |
| W5209 | Single inhumation | -2.9 | 0.70857 |
| W2463 | Single inhumation | -2.8 | 0.70869 |
| W2472 | Single inhumation | -2.8 | 0.70867 |
| W1896 | Single inhumation | -5.9 | 0.70879 |
| W6049 | Single inhumation | -5.4 | 0.70867 |
| W1901 | Single inhumation | -5.1 | 0.70867 |
| W6083 | Single inhumation | -4.5 | 0.70849 |
| W2485 | Single inhumation | -4.2 | 0.70862 |
| W6111 | Single inhumation | -4.0 | 0.70862 |
| W4324 | Single inhumation | -4.0 | 0.70911 |
| W2301 | Single inhumation | -3.8 | 0.70863 |
| W3612 | Single inhumation | -3.6 | 0.70879 |
| W3702 | Single inhumation | -3.5 | 0.70871 |
| W2468 | Single inhumation | -3.4 | 0.70876 |
| W2831 | Single inhumation | -3.3 | 0.70846 |
| W3182 | Single inhumation | -3.3 | 0.70898 |
| W0303 | Single inhumation | -3.3 | 0.70876 |
| W1838 | Single inhumation | -3.2 | 0.70854 |
| W6112 | Single inhumation | -3.1 | 0.70878 |
| W6022 | Single inhumation | -3.0 | 0.70912 |
| W2499 | Single inhumation | -3.0 | 0.70872 |
| W336 | FC1 | -6.1 | 0.71010 |
| W482 | FC1 | -7.1 | 0.70972 |
| W403 | FC1 | -6.4 | 0.70999 |
| W276 | FC1 | -7.0 | 0.70941 |
| W396 | FC1 | -6.4 | 0.70957 |
| W494 | FC2 | -4.0 | 0.70918 |
| W428 | FC2 | -5.9 | 0.70963 |
| W464 | FC2 | -6.7 | 0.70901 |
| W503 | FC2 | -6.1 | 0.70979 |
| W461 | FC2 | -6.7 | 0.70929 |
| W429 | FC2 | -7.3 | 0.70899 |
| W577 | FC2 | -6.4 | 0.70969 |
| W462 | FC2 | -5.4 | 0.70892 |
| W463 | FC2 | -4.4 | 0.70861 |
| W576 | FC2 | -5.2 | 0.70916 |
| W737 | FC2 | -5.9 | 0.70950 |
| W808 | FC3 | -6.1 | 0.70952 |
| W810 | FC3 | -5.3 | 0.70939 |
| W696 | FC3 | -5.7 | 0.70934 |
| W814 | FC3 | -5.5 | 0.71053 |
| W809 | FC3 | -5.9 | 0.71062 |
| W701 | FC3 | -6.6 | 0.70843 |

(*Continued*)

**Table 3.** (Continued)

| ID | Burial Type | $\delta^{18}$O (VPDB, ‰) | $^{87}$Sr/$^{86}$Sr |
|---|---|---|---|
| W702 | FC3 | -6.9 | 0.70913 |
| W699 | FC3 | -6.6 | 0.70882 |
| W812 | FC3 | -3.8 | 0.70929 |
| W706 | FC3 | -3.9 | 0.70904 |
| W704 | FC3 | -6.6 | 0.71071 |
| W705 | FC3 | -4.5 | 0.70882 |
| W807 | FC3 | -5.1 | 0.70940 |
| W703 | FC3 | -6.2 | 0.70894 |
| W653 | FC3 | -6.5 | 0.70907 |
| W698 | FC3 | -5.2 | 0.70897 |
| W811 | FC3 | -7.1 | 0.70990 |
| W650 | FC3 | -7.0 | 0.71001 |
| W1783 | FC4 | -4.8 | 0.70947 |
| W1781 | FC4 | -6.1 | 0.70951 |
| W1770 | FC4 | -5.4 | 0.70893 |
| W1777 | FC4 | -5.6 | 0.70906 |
| W1773 | FC4 | -6.1 | 0.70945 |
| W1779 | FC4 | -5.7 | 0.70927 |
| W1771 | FC4 | -5.0 | 0.70888 |
| W1774 | FC4 | -5.3 | 0.70992 |
| W2588 | FC5 | -4.7 | 0.70849 |
| W2587 | FC5 | -4.5 | 0.70853 |
| W2589 | FC5 | -4.5 | 0.70855 |
| W2590 | FC5 | -4.3 | 0.70856 |
| W2737 | FC6 | -4.8 | 0.70873 |
| W2738 | FC6 | -3.4 | 0.70885 |
| W2739 | FC6 | -3.9 | 0.70898 |
| W2825 | FC7 | -5.0 | 0.70811 |
| W2764 | FC7 | -3.8 | 0.70834 |
| W4378 | FC8 | -4.5 | 0.70882 |
| W4376 | FC8 | -3.8 | 0.70886 |
| W4380 | FC8 | -3.6 | 0.70868 |
| W4674 | FC9 | -4.5 | 0.70888 |
| W4666 | FC9 | -4.8 | 0.70877 |
| W4680 | FC9 | -3.5 | 0.70883 |
| W4651 | FC9 | -4.5 | 0.70881 |
| W4670 | FC9 | -4.3 | 0.70873 |
| W4689 | FC9 | -4.2 | 0.70881 |
| W4684 | FC9 | -3.7 | 0.70897 |
| W4687 | FC9 | -3.7 | 0.70911 |

FC refers to the mass grave (*fossa comune*) in which individuals were buried.

## 409 BCE

The 11 individuals from the 409 BCE mass graves exhibit a mean $^{87}$Sr/$^{86}$Sr value of 0.70884±
0.00012. Their strontium values are not significantly different from the general populace (W =
280, p-value = 0.01621). One individual (9% of sample) falls outside the local $^{87}$Sr/$^{86}$Sr range.

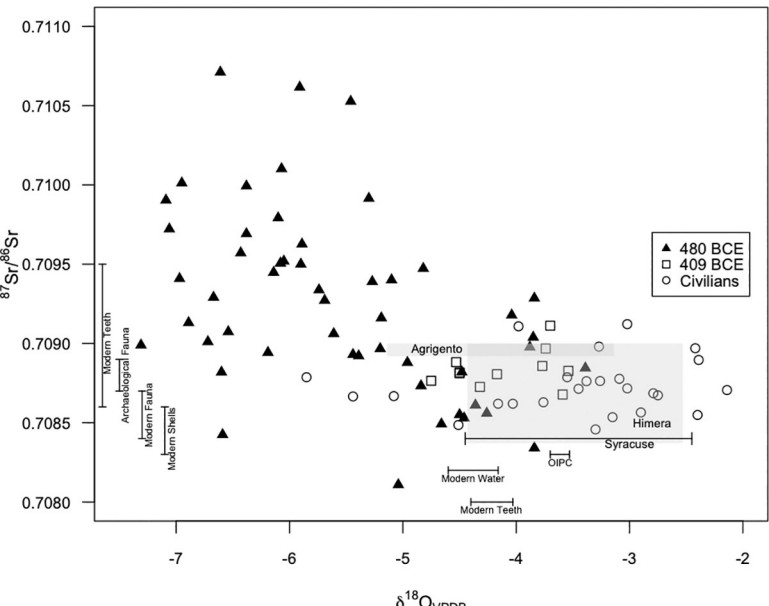

**Fig 4. $^{87}$Sr/$^{86}$Sr and $\delta^{18}$O data for soldiers from 480 BCE and 409 BCE, and civilians.** Gray shaded areas represent the measured baseline values for Agrigento and Himera. The bar representing Syracuse's expected $\delta^{18}$O values is placed close to the $^{87}$Sr/$^{86}$Sr values that can be expected for Syracuse, based on its underlying geology (0.70800–0.70900). Bars at the base of the figure represent mean/stdev $\delta^{18}$O values of measured modern water, measured modern enamel, and estimated precipitation at Himera. Bars on the left of the figure represent mean/stdev $^{87}$Sr/$^{86}$Sr values of measured modern teeth and fauna, archaeological fauna, and shells at Himera.

The 409 BCE individuals exhibit a mean $\delta^{18}$O value of -4.1±0.4‰. Their $\delta^{18}$O values are significantly different from those of the general populace (W = 62, p-value = 0.009996). Thirty-six percent of soldiers from 409 BCE are outside the civilian $\delta^{18}$O range. Soldiers from 480 BCE and soldiers from 409 BCE exhibit significantly different $^{87}$Sr/$^{86}$Sr and $\delta^{18}$O values (W = 433, p-value = 0.005098; W = 62, p-value = 5.9x10$^{-5}$ respectively).

## Discussion

The variability in isotope ratios of armed forces exceeds that of the general populace at Himera, corroborating historical texts that described an alliance of Greek soldiers from other regions. Most of the individuals in mass graves associated with the 480 BCE battle exhibit non-local isotopic values, supporting the interpretation that soldiers from elsewhere aided Himera in 480 BCE. The presence of a high number of non-locals in the earlier battle is especially interesting given recent research that overall connectivity and migration in the Mediterranean was relatively low according to measured $^{87}$Sr/$^{86}$Sr values of individuals from several sites in the region [65]. Most of the individuals in mass graves of the 409 BCE battle exhibit local isotopic values which supports literary accounts that a mostly local force of soldiers defended Himera when it lost in 409 BCE.

This study provides evidence, however, for an aspect of warfare that was not emphasized by ancient historians: the presence of mercenaries from beyond Sicily among the soldiers defending Himera in 480 BCE. The range and large variance in strontium and oxygen isotope values observed among soldiers from the 480 BCE battle exceeds the baseline ranges of values of Himera, Agrigento, and Syracuse alike. The comparatively high $^{87}$Sr/$^{86}$Sr values of non-local soldiers point to regions with more ancient underlying rock formations; similarly high $^{87}$Sr/$^{86}$Sr values are seen in the very ancient Paleozoic granites of the Central Cyclades in the

Aegean or the Triassic sandstones and Paleozoic metamorphic rocks along the Catalan Coastal Range in the north-east Iberian peninsula [45, 66]. Himera, Agrigento, and Syracuse all have similar $\delta^{18}$O values, ranging from approximately -5‰ to -2‰. However, many of the $\delta^{18}$O values from soldiers of 480 BCE fall below all three cities, suggesting they were not Sicilian Greeks. The comparatively low $\delta^{18}$O values of non-local soldiers suggest they originated from regions located further inland, at higher elevations, or at higher latitudes than Sicily. Similarly low values have been reported for Ephesus, Thebes, and Apollonia along the Black Sea [47, 54]. Together, $^{87}$Sr/$^{86}$Sr and $\delta^{18}$O evidence suggest many of the soldiers could have come from elsewhere in the Mediterranean, drawn to Sicily as mercenaries to round out Greek armies. Further baseline sampling across Sicily and mainland Greece is warranted to confirm this departure of the bioarchaeological evidence from ancient literary sources.

## Geopolitics of Sicily: The practices of tyrants and the question of mercenaries

It is not disputed that mercenaries, of Greek and other origins, were used in armies across the Mediterranean (Hdt 8.26.52, 7.165; Thuc.I.60) [13, 15, 57, 58]. Herodotus himself is clear about the presence of mercenaries in the Carthaginian army that attacked Himera (Hdt. 7.165) [67, 68]. What is less clear, in part because of their omission by Herodotus and Diodorus Siculus, is whether Sicilian Greek tyrants hired non-Greek mercenaries to fight alongside citizen armies. Literary sources suggest Hippocrates of Gela hired indigenous Sicels (i.e. Sikels) for his army (Polyaen 5:6). Gelon hired Greek mercenaries from Arcadia after succeeding Hippocrates as tyrant of Gela in the 5th century BCE [13, 69]. While Gelon is not explicitly said to have hired *foreign* mercenaries for his army in 480 BCE, by 466 BCE the tyranny in Syracuse is abolished and Diodorus Siculus mentions that Gelon enfranchised 7,000 of 10,000 foreign mercenaries (*xenosmisthophoros*), allowing them to remain in the city as citizens, despite the displeasure it roused among the rest of its citizenry (Diod. 11:72).

The social migration from mercenary to citizen impacts ideas of citizenship and identity within the *poleis* in Sicily, providing a pathway for "barbarians" or outsiders to the *poleis* to gain access to the rights of citizens [13, 15, 16, 18, 70]. Whereas the hoplite soldier was a symbol for other social changes in the ancient Greek world, such as the rise of the middle class [11] in the Classical period, the hiring of mercenaries might have been seen as their antithesis, despite the increasing popularity of the practice. Terms to describe the paid soldiers were increasingly derogatory in the Classical period ("wage earner" *misthophoros*) to separate them from social classes who saw wages as deprecatory. The shifting allegiances of mercenaries offended ideals of citizenship and loyalty [15]. Ancient writers, in particular Herodotus who was writing during the Classical period, shared these attitudes (Hdt. 8.26.1) [9] and may have downplayed the role of mercenaries in Greek armies. While literary sources overlook mercenaries' role in the Greek victory at Himera in 480 BCE, isotopic evidence suggests tyrants were indeed looking beyond their local or regional communities as they consolidated and maintained power on Sicily, and that foreign mercenaries played a significant role in the fates of Greek colonies.

## Conclusion

Human skeletons from the Battles of Himera offered an unprecedented opportunity to evaluate historical accounts of the Battles recorded by Diodorus Siculus and Herodotus, and to evaluate the composition of Greek armies in the Classical period. Stable isotope ratios from mass graves associated with the Battles of Himera support ancient historians' accounts of a Greek alliance that saved Himera in 480 BCE and promoted Gelon to a celebrated ruler in Sicily. The soldiers aiding Himera in 480 BCE likely included foreign mercenaries, which could have

included those already in Syracuse hired by Gelon. The isotopic data also supports ancient authors' claims that Himera was unaided in 409 BCE, leading to its demise. The wide range of isotopic values of soldiers from 480 BCE points to a key role played by foreign mercenaries in the Greek victory, expanding on the information ancient authors elect to emphasize.

Greek historians were some of the earliest to write down their people's stories and histories. Most written records of similar antiquity have been lost through time, ensuring a permanent, leading role for the ancient sources that remain as sources of knowledge of the past. Because they cannot be replaced, and because they are not only works of art but unique sources of otherwise inaccessible historical details, critically evaluating literary sources for their fidelity using other available evidence is useful and warranted. The present study supports the fidelity of ancient records of the Battles of Himera in documenting the divergent outcomes of battles aided by others versus fought alone. However, the ethnocentric accounts of ancient authors downplay the true heterogeneous nature of the Greek colonies and armies, likely to align the victory at Himera with other prominent Greek victories across the Mediterranean (e.g. Salamis). Foreign mercenaries played an important role in the military prowess of some Greek armies as early as 480 BCE and reflect the diversity of ancient communities in the western Mediterranean.

## Supporting information

**S1 Table. Specimen numbers for archaeological samples.**
(PDF)

## Acknowledgments

We are grateful to the following individuals and institutions for their assistance with this research: Adam Kazmi, Julianne Stamer, Shirley Sheng, Emmy Deng, and the University of Georgia Center for Applied Isotope Studies; the University of Florida Department of Anthropology and the Department of Geological Sciences; Lorenzo Aquilino, Matteo Valentino, Pier Francesco Fabbri, Norma Lonoce, Giorgia Vincenti, and the Soprintendenze di Palermo.

## Author Contributions

**Conceptualization:** Katherine L. Reinberger.

**Data curation:** Katherine L. Reinberger.

**Formal analysis:** Katherine L. Reinberger.

**Funding acquisition:** Katherine L. Reinberger, Laurie J. Reitsema, Britney Kyle, Stefano Vassallo.

**Methodology:** Katherine L. Reinberger, Laurie J. Reitsema, George Kamenov, John Krigbaum.

**Project administration:** Katherine L. Reinberger, Stefano Vassallo.

**Resources:** Laurie J. Reitsema, Britney Kyle, Stefano Vassallo, George Kamenov, John Krigbaum.

**Supervision:** Laurie J. Reitsema.

**Writing – original draft:** Katherine L. Reinberger.

**Writing – review & editing:** Katherine L. Reinberger, Laurie J. Reitsema, Britney Kyle, Stefano Vassallo, George Kamenov, John Krigbaum.

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
