## [Decision Letter · Decision Letter 0]

27 Jan 2021

PONE-D-20-37019

Isotopic evidence for geographic heterogeneity in Ancient Greek military forces

PLOS ONE

Dear Dr. Reinberger,

Thank you for submitting your manuscript to PLOS ONE. After careful consideration, we feel that it has merit but does not fully meet PLOS ONE’s publication criteria as it currently stands. Therefore, we invite you to submit a revised version of the manuscript that addresses the points raised during the review process.

Reviewer 1:

The scientific merits of this paper include the investigation of geographic origins in individuals buried after two Greek battles in the Greek Sicilian colony of Himera using strontium and oxygen isotopes. The dataset is a welcome contribution to the use of isotopic data to infer geographic origins. I think it is particularly important to have examples like this one, where the authors successfully incorporate Greek historical documents with bioarchaeological data in an appropriately contextualized bioarchaeological project.

The research questions, background sections, sample preparation and analysis are appropriate and the interpretations are reasonable and supported by the data. I only have minor suggestions to improve the paper. For example, the convention is usually to have a minimum of three sentences in each paragraph, and following Coplen, the delta symbol should be italicized throughout. Finally, given the accuracy and precision of the oxygen isotope data from the analyzed standards, the data should only be reported to the first decimal place, not the second.

Overall, I feel that the questions and methods addressed in this manuscript are beneficial to the body of literature on isotopic analyses in bioarchaeology. This manuscript was very interesting to read and I am happy recommend the manuscript for publication.

Reviewer 2

Overall, this is a very good study using scientific methods to test "historical" information. Very few strontium isotope analyses have been done yet in this area, however oxygen isotope values are available for many archaeological sites in Italy. One example for Sicily, for Greek site in Syracuse, is Tanasi et al. 2017 in  Science and Technology of Archaeological Research 3(2): 466-477.

Some minor questions just to elaborate on:

p. 3, line 58: was The Histories definitely written in 440 BCE, or “about” then?

p. 7, lines 150-151: here it says “revealed nine mass graves...” but then in the next sentence “Seven graves...and one.” So is it 8 or 9 in total?

p. 9, lines 181-182: Why are “shales and granites” older rocks, and “basalt” a younger rock? Doesn’t this specifically depend on the particular geographic location? The citations provided are not for Sicily or this area in particular.

p. 11, line 232-234: So of 132 total individuals, 70 didn’t have any teeth? (authors state that their sample is 100% of all having at least one tooth)

p. 13 - Table 1: I don’t think it is appropriate to have six decimal places for the Sr isotope ratio, since the values obtained on the standard NBS 987 have std in the 5th decimal place (p. 16, line 304).

p. 16, line 306: so you decided to use regular acetic acid rather than buffered, when the latter has been shown to be more consistent between labs and separate from particle size. Any particular reason?

p. 16, lines 313-315: was a different mass spec used for the water samples, or just a separate input to the same mass spec as the tooth samples?

Minor corrections:

p. 4, line 79: add space before “Greekness”

p. 5, line 108: should be a period after “Fig”

p. 7, line 146: need space after period and before “The”

p. 8, line 174: plural is “radii”, not “radiuses”

p. 9, line 190 & 192: should be a period after “Fig”

p. 9, line 192: Sentence could have better grammar, e.g. “..rocks are associated with the Mt. Etna...”

p. 10, line 202: add space bore “87Sr/86Sr”

p. 10, line 208: spell out “Fig” since it’s part of the sentence

p. 10, line 210: remove the underline under the period and space before “(52)”

p. 10, lines 219-224: add space before “mm” and before “C”, these are abbreviations for separate words (the degree symbol is not an abbreviation, so that stays with the temperature number)

p. 11, line 25: same

p. 11, line 242: change tense of “analyze” to “analyzed”

p. 12, line 246: same

p. 15, lines 285, 290, 301: add space before “C”

p. 15, line 290: remove space in “2 N” to be consistent with what you have elsewhere on the same page

p. 15, line 298: add spaces in both cases where you have “2ml”

p. 16, line 303: edit wording so you don’t have “following using”

p. 16, line 311: make the sentence better by starting “The mean d18O of the Fisher...”

p. 16, line 312: remove the second decimal place from the std dev “0.12"

p. 16, lines 310, 315, 319, 321, 322: the “v” in “vPDB” and “vSMOW should be capitalized

p 17, line 337: ditto (while in line 339 you do have it capitalized)

p. 17, line 334: shorten to 5 decimal places the Sr range

p. 17, line 341: delete “calculated” since it is in the next line of the sentence

p. 18, Table 3: change d18O isotope values to one decimal place; capitalize the “v” in column caption

p. 18, Table 3: change Sr decimal places to 5

p. 21, lines 349-350: be consistent in hyphen size and space before/after

p. 21, lines 349-350 & 356-357: be consistent and use 5 decimal places (not 4 or 6)

p. 22, line 30: again you have just 4 decimal places, is that particularly intentional?

p. 23, line 376: add something like “The dozen...” at the beginning of the sentence to reinform the reader how many individuals the data are based on

p. 23, line 377: you have average of 6 decimal places, std only 5

p. 23, line 380: add space between “d18O” and “value”, and make “18" superscript

p. 23, line 384: add space again

p. 23, line 392: after “values” add “which” and remove “s” from “supports” so it reads “...values which support literary...”

p. 23, line 394: reword beginning of sentence, e.g. to “This study provides evidence, however, for..”

p. 24, line 420: are “Sicels” the same as “Sikels”? Check if you mention either before this page

p. 25, line 443: change “offer” to past tense (offered)

p. 26, line 462: add space after period, before “Foreign”

p. 26, line 468: change semicolon to comma after “Stamer”

p. 28, ref. 1: remove the “Available from: ...”

p. 29, line 550: capitalize second initial for Ezzo, uncapitalize “Of” in journal name

p. 29, line 564: remove space before colon

p. 30, line 572: remove the “Available from: ...”

p. 30, line 585: remove the “Available from: ...”

p. 30, lines 609-10: capitalization of article title inconsistent with others, check the style expected.

p. 31, line 633: remove space after hyphen so that it reads “Look-Up”

p. 31, lines 635-7: remove the “Available from: ...”

Figure 3: the precision of the O isotope ranges should be consistent with decimal places, while given the precision of the measurements no more than 2 decimal places. Also the lower case “v”

We look forward to receiving your revised manuscript.

Kind regards,

Mario Novak

Academic Editor

PLOS ONE

Additional Editor Comments:

Both referees agree that this is a very good paper that needs only some minor reworking to be acceptable for publication. I tend to agree with the reviewers and would suggest a minor revision based on their comments and suggestions.

Journal Requirements:

2. We note that Figures 1-4 in your submission contain map/satellite images which may be copyrighted. All PLOS content is published under the Creative Commons Attribution License (CC BY 4.0), which means that the manuscript, images, and Supporting Information files will be freely available online, and any third party is permitted to access, download, copy, distribute, and use these materials in any way, even commercially, with proper attribution. For these reasons, we cannot publish previously copyrighted maps or satellite images created using proprietary data, such as Google software (Google Maps, Street View, and Earth). For more information, see our copyright guidelines: http://journals.plos.org/plosone/s/licenses-and-copyright.

2.1.    You may seek permission from the original copyright holder of Figures 1-4 to publish the content specifically under the CC BY 4.0 license. 

2.2.    If you are unable to obtain permission from the original copyright holder to publish these figures under the CC BY 4.0 license or if the copyright holder’s requirements are incompatible with the CC BY 4.0 license, please either i) remove the figure or ii) supply a replacement figure that complies with the CC BY 4.0 license. Please check copyright information on all replacement figures and update the figure caption with source information. If applicable, please specify in the figure caption text when a figure is similar but not identical to the original image and is therefore for illustrative purposes only.

Reviewers' comments:

Reviewer's Responses to Questions

**Comments to the Author**

1. Is the manuscript technically sound, and do the data support the conclusions?

Reviewer #1: Yes

Reviewer #2: Yes

2. Has the statistical analysis been performed appropriately and rigorously? 

Reviewer #1: Yes

Reviewer #2: Yes

3. Have the authors made all data underlying the findings in their manuscript fully available?

Reviewer #1: Yes

Reviewer #2: Yes

4. Is the manuscript presented in an intelligible fashion and written in standard English?

Reviewer #1: Yes

Reviewer #2: Yes

5. Review Comments to the Author

Reviewer #1: Review of “Isotopic Evidence for Geographic Heterogeneity in Ancient Greek Military Forces” by Reinberger et al. for PLOS One

The scientific merits of this paper include the investigation of geographic origins in individuals buried after two Greek battles in the Greek Sicilian colony of Himera using strontium and oxygen isotopes. The dataset is a welcome contribution to the use of isotopic data to infer geographic origins. I think it is particularly important to have examples like this one, where the authors successfully incorporate Greek historical documents with bioarchaeological data in an appropriately contextualized bioarchaeological project.

The research questions, background sections, sample preparation and analysis are appropriate and the interpretations are reasonable and supported by the data. I only have minor suggestions to improve the paper. For example, the convention is usually to have a minimum of three sentences in each paragraph, and following Coplen, the delta symbol should be italicized throughout. Finally, given the accuracy and precision of the oxygen isotope data from the analyzed standards, the data should only be reported to the first decimal place, not the second.

Overall, I feel that the questions and methods addressed in this manuscript are beneficial to the body of literature on isotopic analyses in bioarchaeology. This manuscript was very interesting to read and I am happy recommend the manuscript for publication.

Reviewer #2: Overall, this is a very good study using scientific methods to test "historical" information. Very few strontium isotope analyses have been done yet in this area, however oxygen isotope values are available for many archaeological sites in Italy. One example for Sicily, for Greek site in Syracuse, is Tanasi et al. 2017 in Science and Technology of Archaeological Research 3(2): 466-477.

Some minor questions just to elaborate on:

p. 3, line 58: was The Histories definitely written in 440 BCE, or “about” then?

p. 7, lines 150-151: here it says “revealed nine mass graves...” but then in the next sentence “Seven graves...and one.” So is it 8 or 9 in total?

p. 9, lines 181-182: Why are “shales and granites” older rocks, and “basalt” a younger rock? Doesn’t this specifically depend on the particular geographic location? The citations provided are not for Sicily or this area in particular.

p. 11, line 232-234: So of 132 total individuals, 70 didn’t have any teeth? (authors state that their sample is 100% of all having at least one tooth)

p. 13 - Table 1: I don’t think it is appropriate to have six decimal places for the Sr isotope ratio, since the values obtained on the standard NBS 987 have std in the 5th decimal place (p. 16, line 304).

p. 16, line 306: so you decided to use regular acetic acid rather than buffered, when the latter has been shown to be more consistent between labs and separate from particle size. Any particular reason?

p. 16, lines 313-315: was a different mass spec used for the water samples, or just a separate input to the same mass spec as the tooth samples?

Minor corrections:

p. 4, line 79: add space before “Greekness”

p. 5, line 108: should be a period after “Fig”

p. 7, line 146: need space after period and before “The”

p. 8, line 174: plural is “radii”, not “radiuses”

p. 9, line 190 & 192: should be a period after “Fig”

p. 9, line 192: Sentence could have better grammar, e.g. “..rocks are associated with the Mt. Etna...”

p. 10, line 202: add space bore “87Sr/86Sr”

p. 10, line 208: spell out “Fig” since it’s part of the sentence

p. 10, line 210: remove the underline under the period and space before “(52)”

p. 10, lines 219-224: add space before “mm” and before “C”, these are abbreviations for separate words (the degree symbol is not an abbreviation, so that stays with the temperature number)

p. 11, line 25: same

p. 11, line 242: change tense of “analyze” to “analyzed”

p. 12, line 246: same

p. 15, lines 285, 290, 301: add space before “C”

p. 15, line 290: remove space in “2 N” to be consistent with what you have elsewhere on the same page

p. 15, line 298: add spaces in both cases where you have “2ml”

p. 16, line 303: edit wording so you don’t have “following using”

p. 16, line 311: make the sentence better by starting “The mean d18O of the Fisher...”

p. 16, line 312: remove the second decimal place from the std dev “0.12"

p. 16, lines 310, 315, 319, 321, 322: the “v” in “vPDB” and “vSMOW should be capitalized

p 17, line 337: ditto (while in line 339 you do have it capitalized)

p. 17, line 334: shorten to 5 decimal places the Sr range

p. 17, line 341: delete “calculated” since it is in the next line of the sentence

p. 18, Table 3: change d18O isotope values to one decimal place; capitalize the “v” in column caption

p. 18, Table 3: change Sr decimal places to 5

p. 21, lines 349-350: be consistent in hyphen size and space before/after

p. 21, lines 349-350 & 356-357: be consistent and use 5 decimal places (not 4 or 6)

p. 22, line 30: again you have just 4 decimal places, is that particularly intentional?

p. 23, line 376: add something like “The dozen...” at the beginning of the sentence to reinform the reader how many individuals the data are based on

p. 23, line 377: you have average of 6 decimal places, std only 5

p. 23, line 380: add space between “d18O” and “value”, and make “18" superscript

p. 23, line 384: add space again

p. 23, line 392: after “values” add “which” and remove “s” from “supports” so it reads “...values which support literary...”

p. 23, line 394: reword beginning of sentence, e.g. to “This study provides evidence, however, for..”

p. 24, line 420: are “Sicels” the same as “Sikels”? Check if you mention either before this page

p. 25, line 443: change “offer” to past tense (offered)

p. 26, line 462: add space after period, before “Foreign”

p. 26, line 468: change semicolon to comma after “Stamer”

p. 28, ref. 1: remove the “Available from: ...”

p. 29, line 550: capitalize second initial for Ezzo, uncapitalize “Of” in journal name

p. 29, line 564: remove space before colon

p. 30, line 572: remove the “Available from: ...”

p. 30, line 585: remove the “Available from: ...”

p. 30, lines 609-10: capitalization of article title inconsistent with others, check the style expected.

p. 31, line 633: remove space after hyphen so that it reads “Look-Up”

p. 31, lines 635-7: remove the “Available from: ...”

Figure 3: the precision of the O isotope ranges should be consistent with decimal places, while given the precision of the measurements no more than 2 decimal places. Also the lower case “v”

6. PLOS authors have the option to publish the peer review history of their article (what does this mean?). If published, this will include your full peer review and any attached files.

Reviewer #1: No

Reviewer #2: No

---

## [Author Response · Author response to Decision Letter 0]

1 Mar 2021

The authors are appreciative of the thorough comments by the reviewers. The following document includes the original comments by the reviewers and editor followed by corresponding responses by the author. Additionally, during revisions, the authors discovered a discrepancy between the text and Figure 4 in the baseline Sr range for Agrigento. We have fixed the range in Figure 4 to be consistent with the calculated range in the text. Finally, we included a citation in the discussion section to a new publication about 87Sr/86Sr variation in the Mediterranean region: 

Leppard TP, Esposito C, Esposito M. The bioarchaeology of migration in the ancient 

Mediterranean: Meta-analysis of radiogenic (87Sr/86Sr ) isotope ratios. Journal of Mediterranean 

Archaeology. 2020;2: 211–41.

Reviewer Comments and Author Responses

Reviewer 1:

The scientific merits of this paper include the investigation of geographic origins in individuals buried after two Greek battles in the Greek Sicilian colony of Himera using strontium and oxygen isotopes. The dataset is a welcome contribution to the use of isotopic data to infer geographic origins. I think it is particularly important to have examples like this one, where the authors successfully incorporate Greek historical documents with bioarchaeological data in an appropriately contextualized bioarchaeological project.

The research questions, background sections, sample preparation and analysis are appropriate and the interpretations are reasonable and supported by the data. I only have minor suggestions to improve the paper. For example, the convention is usually to have a minimum of three sentences in each paragraph, and following Coplen, the delta symbol should be italicized throughout (Author response: DONE). Finally, given the accuracy and precision of the oxygen isotope data from the analyzed standards, the data should only be reported to the first decimal place, not the second (Author response: DONE).

Overall, I feel that the questions and methods addressed in this manuscript are beneficial to the body of literature on isotopic analyses in bioarchaeology. This manuscript was very interesting to read and I am happy recommend the manuscript for publication. 

Reviewer 2

Overall, this is a very good study using scientific methods to test "historical" information. Very few strontium isotope analyses have been done yet in this area, however oxygen isotope values are available for many archaeological sites in Italy. One example for Sicily, for Greek site in Syracuse, is Tanasi et al. 2017 in Science and Technology of Archaeological Research 3(2): 466-477.

Some minor questions just to elaborate on:

p. 3, line 58: was The Histories definitely written in 440 BCE, or “about” then? (Author response: Changed to “around” after further research showing some sources say 440, others say 430, and others say “published between 426 and 415”)

p. 7, lines 150-151: here it says “revealed nine mass graves...” but then in the next sentence “Seven graves...and one.” So is it 8 or 9 in total? (Author response: changed in manuscript to “revealed eight mass graves” and clarified in the text that during excavation they were originally labeled as two separate because of rain damage, but later interpretations indicate they were likely one grave)

p. 9, lines 181-182: Why are “shales and granites” older rocks, and “basalt” a younger rock? Doesn’t this specifically depend on the particular geographic location? The citations provided are not for Sicily or this area in particular. (Author response: Yes, we agree with this point. We removed “older” and “younger” and modified the text accordingly.)

p. 11, line 232-234: So of 132 total individuals, 70 didn’t have any teeth? (authors state that their sample is 100% of all having at least one tooth) (Author response: 25 individuals did not have teeth, 45 individuals were poorly preserved and comingled so matching the bones and teeth of individuals was not possible. Have clarified in the text that due to poor preservation this is the complete, available sample of teeth.) 

p. 13 - Table 1: I don’t think it is appropriate to have six decimal places for the Sr isotope ratio, since the values obtained on the standard NBS 987 have std in the 5th decimal place (p. 16, line 304). (Author response: all Sr isotope ratio values were changed to only report to the 5th decimal place)

p. 16, line 306: so you decided to use regular acetic acid rather than buffered, when the latter has been shown to be more consistent between labs and separate from particle size. Any particular reason?

(Author response: Previous research has shown that both 1.0 M buffered acetic acid, and 0.1 M acetic acid, are effective at removing organic matter and diagenetic carbonates while minimizing dissolution or alteration biogenic of hydroxyapatite (Koch et al. 1997; Garvie-Lok et al. 2004).)

p. 16, lines 313-315: was a different mass spec used for the water samples, or just a separate input to the same mass spec as the tooth samples?

(Author response: A different mass spec was used for the water samples. Both mass specs were Thermo Delta V plus IRMS. The water samples are routed from the TC-EA to the Delta V via a Conflo III interface.)

Minor corrections:

p. 4, line 79: add space before “Greekness” (Author response: DONE)

p. 5, line 108: should be a period after “Fig” (PLOS One style guide requires in-text citations of figures written as “Fig 1” without period)

p. 7, line 146: need space after period and before “The” (Author response: DONE)

p. 8, line 174: plural is “radii”, not “radiuses” (Author response: DONE)

p. 9, line 190 & 192: should be a period after “Fig” (Author response: PLOS One style guide requires in-text citations of figures written as “Fig 1” without period)

p. 9, line 192: Sentence could have better grammar, e.g. “..rocks are associated with the Mt. Etna...” (AUTHOR RESPONSE: DONE)

p. 10, line 202: add space bore “87Sr/86Sr” (Author response: DONE)

p. 10, line 208: spell out “Fig” since it’s part of the sentence (Author response: DONE)

p. 10, line 210: remove the underline under the period and space before “(52)” (Author response: DONE)

p. 10, lines 219-224: add space before “mm” and before “C”, these are abbreviations for separate words (the degree symbol is not an abbreviation, so that stays with the temperature number) (Author response: DONE)

p. 11, line 25: same (Author response: DONE)

p. 11, line 242: change tense of “analyze” to “analyzed” (Author response: DONE)

p. 12, line 246: same (Author response: DONE)

p. 15, lines 285, 290, 301: add space before “C” (Author response: DONE)

p. 15, line 290: remove space in “2 N” to be consistent with what you have elsewhere on the same page (Author response: DONE)

p. 15, line 298: add spaces in both cases where you have “2ml” (Author response: DONE)

p. 16, line 303: edit wording so you don’t have “following using” (Author response: DONE)

p. 16, line 311: make the sentence better by starting “The mean d18O of the Fisher...” (Author response: DONE)

p. 16, line 312: remove the second decimal place from the std dev “0.12" (Author response: DONE)

p. 16, lines 310, 315, 319, 321, 322: the “v” in “vPDB” and “vSMOW should be capitalized (Author response: DONE)

p 17, line 337: ditto (while in line 339 you do have it capitalized) (Author response: DONE)

p. 17, line 334: shorten to 5 decimal places the Sr range (Author response: DONE)

p. 17, line 341: delete “calculated” since it is in the next line of the sentence (Author response: DONE)

p. 18, Table 3: change d18O isotope values to one decimal place; capitalize the “v” in column caption (Author response: DONE)

p. 18, Table 3: change Sr decimal places to 5 (Author response: DONE)

p. 21, lines 349-350: be consistent in hyphen size and space before/after (Author response: DONE)

p. 21, lines 349-350 & 356-357: be consistent and use 5 decimal places (not 4 or 6) (Author response: DONE)

p. 22, line 30: again you have just 4 decimal places, is that particularly intentional? (Author response: originally wanted to communicate less precision because range for Syracuse is based on local geology estimations while Agrigento was based on baseline samples collected. Have added another decimal place to be more consistent with rest of paper)

p. 23, line 376: add something like “The dozen...” at the beginning of the sentence to reinform the reader how many individuals the data are based on (Author response: DONE)

p. 23, line 377: you have average of 6 decimal places, std only 5 (Author response: DONE)

p. 23, line 380: add space between “d18O” and “value”, and make “18" superscript (Author response: DONE)

p. 23, line 384: add space again (Author response: DONE)

p. 23, line 392: after “values” add “which” and remove “s” from “supports” so it reads “...values which support literary...” (Author response: DONE)

p. 23, line 394: reword beginning of sentence, e.g. to “This study provides evidence, however, for..” (Author response: DONE)

p. 24, line 420: are “Sicels” the same as “Sikels”? Check if you mention either before this page (Author response: they are the same, just different spellings, added “Sicels (i.e. Sikels)” to clarify)

p. 25, line 443: change “offer” to past tense (offered) (Author response: DONE)

p. 26, line 462: add space after period, before “Foreign” (Author response: DONE)

p. 26, line 468: change semicolon to comma after “Stamer” (Author response: DONE)

p. 28, ref. 1: remove the “Available from: ...” (Author response: DONE)

p. 29, line 550: capitalize second initial for Ezzo, uncapitalize “Of” in journal name (Author response: DONE)

p. 29, line 564: remove space before colon (Author response: DONE)

p. 30, line 572: remove the “Available from: ...” (Author response: DONE)

p. 30, line 585: remove the “Available from: ...” (Author response: DONE)

p. 30, lines 609-10: capitalization of article title inconsistent with others, check the style expected. (Author response: DONE)

p. 31, line 633: remove space after hyphen so that it reads “Look-Up” (Author response: DONE)

p. 31, lines 635-7: remove the “Available from: ...” (Author response: DONE)

Figure 3: the precision of the O isotope ranges should be consistent with decimal places, while given the precision of the measurements no more than 2 decimal places. Also the lower case “v” (Author response: DONE)

Author response to journal requirements:

1. Please ensure that your manuscript meets PLOS ONE's style requirements, including those for file naming. (Author response: DONE)

2. We note that Figures 1-4 in your submission contain map/satellite images which may be copyrighted. 

Author response:

Figure 1 was edited to remove copyrighted images. The original Mediterranean Sea image (from Google Maps) was replaced with a public domain image from the Wikimedia Commons (https://commons.wikimedia.org/wiki/File:Mediterranean_Sea_16.61811E_38.99124N.jpg), taken from NASA’s globe software World Wind. The plan of Himera was replaced with a new image of Himera created by author SV. The base map of sites in Sicily was created in QGIS by author KLR.

Captions of Figures 2 and 3 were modified to include the data sources used to create the images in QGIS by author KLR. 

Figure 4 is a scatter plot with original data that does not contain map/satellite images so was not modified.

---

## [Editor Report · Decision Letter 1]

8 Mar 2021

Isotopic evidence for geographic heterogeneity in Ancient Greek military forces

PONE-D-20-37019R1

Dear Dr. Reinberger,

We’re pleased to inform you that your manuscript has been judged scientifically suitable for publication and will be formally accepted for publication once it meets all outstanding technical requirements.

Kind regards,

Mario Novak

Academic Editor

PLOS ONE
---

## [Editor Report · Acceptance letter]

18 Mar 2021

PONE-D-20-37019R1 

Isotopic evidence for geographic heterogeneity in Ancient Greek military forces 

Dear Dr. Reinberger:

I'm pleased to inform you that your manuscript has been deemed suitable for publication in PLOS ONE. Congratulations! Your manuscript is now with our production department. 

Kind regards, 

on behalf of

Dr. Mario Novak 

Academic Editor

PLOS ONE